# TGF-β Isoforms and Local Environments Greatly Modulate Biological Nature of Human Retinal Pigment Epithelium Cells

**DOI:** 10.3390/bioengineering11060581

**Published:** 2024-06-07

**Authors:** Nami Nishikiori, Tatsuya Sato, Toshifumi Ogawa, Megumi Higashide, Araya Umetsu, Soma Suzuki, Masato Furuhashi, Hiroshi Ohguro, Megumi Watanabe

**Affiliations:** 1Departments of Ophthalmology, Sapporo Medical University, S1W17, Chuo-ku, Sapporo 060-8556, Japan; nami076@yahoo.co.jp (N.N.); megumi.h@sapmed.ac.jp (M.H.); araya.alaya.favreweissth@gmail.com (A.U.); ophthalsoma@sapmed.ac.jp (S.S.); ooguro@sapmed.ac.jp (H.O.); 2Departments of Cardiovascular, Renal and Metabolic Medicine, Sapporo Medical University, S1W17, Chuo-ku, Sapporo 060-8556, Japan; sato.tatsuya@sapmed.ac.jp (T.S.); a08m024@yahoo.co.jp (T.O.); furuhasi@sapmed.ac.jp (M.F.); 3Departments of Cellular Physiology and Signal Transduction, Sapporo Medical University, S1W17, Chuo-ku, Sapporo 060-8556, Japan

**Keywords:** TGF-β isoforms, human retinal pigment epithelium, 3D culture, hypoxia, seahorse cellular metabolic functions

## Abstract

To characterize transforming growth factor-β (TGF-β) isoform (TGF-β1~3)-b’s biological effects on the human retinal pigment epithelium (RPE) under normoxia and hypoxia conditions, ARPE19 cells cultured by 2D (two-dimensional) and 3D (three-dimensional) conditions were subjected to various analyses, including (1) an analysis of barrier function by trans-epithelial electrical resistance (TEER) measurements; (2) qPCR analysis of major ECM molecules including *collagen 1 (COL1)*, *COL4*, and *COL6*; *α-smooth muscle actin (αSMA)*; *hypoxia-inducible factor 1α (HIF1α)*; and *peroxisome proliferator-activated receptor-gamma coactivator (PGC1α)*, a master regulator for mitochondrial respiration;, tight junction-related molecules, *Zonula occludens-1 (ZO1)* and *E-cadherin*; and *vascular endothelial growth factor (VEGF)*; (3) physical property measurements of 3D spheroids; and (4) cellular metabolic analysis. Diverse effects among TGF-β isoforms were observed, and those effects were also different between normoxia and hypoxia conditions: (1) TGF-β1 and TGF-β3 caused a marked increase in TEER values, and TGF-β2 caused a substantial increase in TEER values under normoxia conditions and hypoxia conditions, respectively; (2) the results of qPCR analysis supported data obtained by TEER; (3) 3D spheroid sizes were decreased by TGF-β isoforms, among which TGF-β1 had the most potent effect under both oxygen conditions; (4) 3D spheroid stiffness was increased by TGF-β2 and TGF-β3 or by TGF-β1 and TGF-β3 under normoxia conditions and hypoxia conditions, respectively; and (5) the TGF-β isoform altered mitochondrial and glycolytic functions differently under oxygen conditions and/or culture conditions. These collective findings indicate that the TGF-β-induced biological effects of 2D and 3D cultures of ARPE19 cells were substantially diverse depending on the three TGF-β isoforms and oxygen levels, suggesting that pathological conditions including epithelial–mesenchymal transition (EMT) of the RPE may be exclusively modulated by both factors.

## 1. Introduction

Human retinal pigment epithelium (RPE) is a major component of the outer blood–retinal barrier (oBRB), which helps to maintain physiological homeostasis of the retinal outer segments including the choroid [1]. In addition, the RPE is a pivotal segment in the retinal pathogenesis of neovascularization and fibrogenesis, including age-related macular degeneration (ARMD) and proliferative vitreoretinopathy (PVR) [2]. As a possible factor underlying those pathogenic mechanisms, epithelial–mesenchymal transition (EMT)-related fibrogenesis of the RPE is primarily involved. It is also thought that EMT of the RPE can induce a break in the BRB that results in the development of choroidal neovascularization (CNV) [3]. Since it has been shown that vitreous levels of transforming growth factor-beta 2 (TGF-β2) are increased in patients with PVR [4,5,6,7], TGF-β isoforms, in addition to their receptors, are also identified in surgical specimens from patients with PVR [3,8,9,10]. TGF-β2 is known as one of the potent inducers of pathological EMT.

Hypoxia has also been suggested to be a pathogenic factor in the development of CNV [11]. In fact, hypoxia facilitates the secretion of various proangiogenic cytokines, such as erythropoietin (EPO) and vascular endothelial growth factor (VEGF), which are induced by a mechanism related to hypoxia-inducible factors (HIFs). They are thereby involved in the development of several retinal diseases, including diabetic retinopathy (DR), retinal vein occlusion (RVO), and ARMD, in addition to the retinopathy of prematurity (ROP) [12,13,14]. Since it was shown that TGF-β2-related EMT of the RPE caused dysfunctions of mitochondrial respiration [15], TGF-β2-related EMT of RPE cells and oxygen conditions may be closely linked. It is extremely important to understand in more detail the underlying mechanisms responsible for retinal neovascularization and fibrogenesis. Recently, to study the effects of both factors on EMT of the RPE, TGF-β2-related EMT was studied under normoxia and hypoxia conditions using a three-dimensional (3D) culture [16,17,18,19] of the human RPE cell line, ARPE19 cells, in addition to their two-dimensional (2D) culture. The results showed significant alterations of TGF-β2-related EMT in both the 2D- and 3D-cultured ARPE19 cells caused by different oxygen conditions, despite the cells’ different metabolic plasticity. Therefore, these observations suggested that EMT of ARPE19 cells may be significantly modulated during their spatial spreading changes [20].

Among members of the human TGF-β family, TGF-β-1, TGF-β-2, TGF-β-3, and others have been identified [21]. Functionally, these three TGF-β isoforms induce different biological functions by binding the identical receptors through the Smad2/3 signaling mechanism [22,23,24,25]. Intraocularly, elevated levels of these three TGF-β isoforms in the aqueous humor (AH) have been shown [26], and these isoforms are involved in different ways in glaucoma pathogenesis [27,28,29]. In a previous study, TGF-β1, TGF-β2, and TGF-β3 were localized differently in the posterior segments of human eyes [30], and gene expression of the three receptors, TGFBR1, TGFBR2, and TGFBR3, was also identified in the chick retina, RPE, and choroid [31]. On the other hand, it was reported that the expression of TGF-β1 and -β2 was only detected in cultured human RPE cells [32]. Collectively, these findings suggested that the TGF-β isoforms have different physiological and pathological roles, including EMT of the RPE, though their roles have not been elucidated.

Here, to compare the induced biological nature of the three TGF-β isoforms under different oxygen conditions (normoxia and hypoxia), including EMT of the RPE, 2D- and 3D-cultured ARPE19 cells were prepared and subjected to barrier function analysis using trans-epithelial electrical resistance (TEER) measurements (2D), seahorse mitochondrial and glycolytic function analysis (2D), physical property analysis of 3D spheroids, and (4) qPCR analysis of various factors related to ECM, EMT, hypoxia, and cell junction.

## 2. Materials and Methods

Two-dimensional and three-dimensional cultures of human retinal pigment epithelium (ARPE19) cells

To use a commercially available human retinal pigment epithelium cell line, ARPE-19 (ATCC, #CRL-2302™), we complied with the tenets of the Declaration of Helsinki, and all experimental protocols were approval by the internal review board of Sapporo Medical University.

Two- and three-dimensional cell cultures were maintained as in recently described methods [20]. Briefly, under standard normoxia conditions (37 °C, 5% CO_2_) or hypoxia conditions (37 °C, 5% CO_2_, 1% O_2_), ARPE19 cells were cultured in 2D 150 mm culture dishes in HG-DMEM medium supplemented with 10% FBS, 1% L-glutamine, and 1% antibiotic-antimycotic. They were maintained by changing the medium every other day. Alternatively, 3D ARPE19 spheroid culture was processed in a hanging drop culture plate (#HDP1385, Sigma-Aldrich, St. Louis, MO, USA) using the above 2D culture medium supplemented with 0.25% methylcellulose (Methocel A4M) and maintained by changing half of the medium every other day until Day 6. On Day 1, 5 ng/mL TGF-β1~3 was administered to the medium to stimulate EMT.

Measurement of TEER

As described in previous reports [19,33], 2D-cultured ARPE19 cells were cultured in a TEER plate (0.4 μm pore and 12 mm diameter; Corning Transwell, Sigma-Aldrich, St. Louis, MO, USA), and TEER values were measured using an electrical resistance device (KANTO CHEMICAL Co., Inc., Tokyo, Japan).

Measurement of the size and solidity of 3D ARPE19 spheroids

To characterize the physical aspects of 3D ARPE19 spheroids, their mean size (μm^2^) and stiffness (μN/μm, requiring force for compression until their semi-diameter could be measured during a period of 20 s) were evaluated by measuring the largest cross-sectional area (CSA) of phase-contrast images obtained using an inverted microsxfcope (Nikon ECLIPSE TS2; Tokyo, Japan) and direct compression using a micro-squeezer (MicroSquisher, CellScale, Waterloo, ON, Canada), respectively, as reported in a previous study [16,18].

Measurement of real-time mitochondrial and glycolytic functions

For mitochondrial and glycolytic functions, the oxygen consumption rates (OCR) and the extracellular acidification rates (ECAR) of 2D- and 3D-cultured ARPE19 cells were determined using a Seahorse XFe96 Bioanalyzer (Agilent Technologies, Santa Clara, CA, USA) according to the manufactur er’s protocols.

In brief, on the day before the assay, the Seahorse cartridge was hydrated overnight with ddH_2_O at 37 °C in a CO_2_-free incubator. For 2D cells, approximately 20,000 cells were placed in each well of an XFe96 Cell Culture Microplate (Agilent Technologies, #103794-100) 24 h before the assay. The culture medium was then replaced with 180 μL of Seahorse XF DMEM assay medium (Agilent Technologies, #103575-100) supplemented with 5.5 mM glucose, 2.0 mM glutamine, and 1.0 mM sodium pyruvate for the Seahorse XF Cell Mito Stress Test or with only 2.0 mM glutamine for the XF Glycolysis Stress Test, respectively, just before the assay. For 3D-cultured cells, on the day of the assay, one to three spheroids, depending on the size, were transferred into each well of an XFe96 Spheroid Microplate (Agilent Technologies, #102978-100) that was pre-coated with Cell-Tak (Corning #354240), and the plate was filled with 180 μL of each Seahorse assay buffer as described above, depending on the type of assay.

For the Seahorse XF Cell Mito Stress Test, the values of OCR were determined in a 3 min mix, 3 min measure protocol at baseline and with the following sequential injections: oligomycin (final concentration: 2.0 μM), carbonyl cyanide p-trifluoromethoxyphenylhydrazone (FCCP, final concentration: 5.0 μM), and rotenone/antimycin A mixture (final concentration: 1.0 μM). For the Seahorse XF Glycolytic Stress Test, the values of ECAR were determined in a 3 min mix, 3 min measure protocol at baseline and with the following sequential injections: glucose until 5.5 mM, oligomycin until 2.0 μM, and 2-deoxyglucose until 50 mM. Because of the difference in the effects of injected drugs between 2D cells and 3D spheroids, 3 cycles of each measurement, 4 cycles at baseline, and 6 cycles with each injected drug were employed as comparable analyses for the 2D cells and the 3D spheroids. For the measurement of hypoxic conditions, the Seahorse bioanalyzer instrument was placed in a hypoxic chamber with ~3% O_2_ concentration, and the OCR and ECAR were determined. After assays were complete, the values for OCR and ECAR were normalized by the amount of protein of the 2D cells determined by a BCA protein assay (TaKaRa Bio, Shiga, Japan), or those values were normalized by the number of 3D spheroids, respectively.

Other analytical methods

Using specific primers and probes, as shown in Appendix A, qPCR analysis of the 2D or 3D cultures of ARPE19 cells was performed as previously reported [17,34]. Briefly, total RNA was extracted using an RNeasy mini kit (Qiagen, Valencia, CA, USA), and reverse transcription was processed with the SuperScript IV kit (Invitrogen, Tokyo, Japan). Respective gene expression was quantified by real-time PCR (Cq value = 40) with Taqman Master mix (Applied Biosystems/Thermo Fisher Scientific, Waltham, MA, USA). cDNA quantities were normalized to the expression of the housekeeping gene *36B4* (*Rplp0*, IDT, San Jose, CA, USA) and are shown as fold changes relative to the control. To perform statistical analyses of experimental data, Graph Pad Prism, version, 10.2.2 (397) software (San Diego, CA, USA) was used as described in previous reports [17,34].

## 3. Results

To compare the EMT of the RPEs under different oxygen conditions caused by three major TGF-β isoforms, TGF-β1~3, we confirmed in advance no difference in cell viabilities in the 2D- and 3D-cultured ARPE19 cells by the administration of 5 ng/mL TGF-β1~3 under 20% O_2_ (normoxia) or 1% O_2_ (hypoxia) conditions, as observed in our preceding study [20]. Initially, to estimate the barrier function of the RPE, as a simple in vitro model for the outer Blood Retinal Barrier (oBRB), 2D ARPE19 monolayers were subjected to a TEER measurement. Their culture conditions were confirmed by phase-contrast microscope (Figure 1 right panels). As shown in Figure 1 panel A, a TGF-β2-induced increase in the TEER value was observed under hypoxia conditions but not under normoxia conditions, as consistently observed in our preceding study [20]. In contrast, TGF-β1 and TGF-β3 had different effects, that is, they both substantially increased the TEER value under normoxia conditions but not under hypoxia conditions.

Next, levels of mRNA expression of various molecules including major ECM proteins *collagen 1 (COL1)*, *COL4*, *and COL6; α-smooth muscle actin (αSMA); hypoxia-inducible factor 1α (HIF1α)*; and *peroxisome proliferator-activated receptor-gamma coactivator (PGC1α)*, a master regulator for mitochondrial respiration; tight junction-related molecules *Zonula occludens-1 (ZO1)* and *E-cadherin*; and *vascular endothelial growth factor (VEGF)* were compared among conditions as described above. As shown in Figure 2, despite the fact that mRNA expression of *COL1* and *COL4* was not altered by TGF-β3, their expression was substantially upregulated by TGF-β1 and TGF-β2, and those increased levels of TGF-β1 were much more potent under both normoxia and hypoxia conditions. The gene expression of *αSMA* was also substantially upregulated by TGF-β1 compared to the expression of other genes under both oxygen conditions. In contrast, the mRNA expression of COL6 was markedly upregulated by TGF-β3 under both normoxia and hypoxia conditions. As for the mRNA expression levels of tight junction-related molecules, (1) the mRNA expression of *ZO1* and *E-cadherin* was significantly downregulated by hypoxia conditions, (2) that of *ZO1* was substantially upregulated by TGF-β1 and TGF-β3 under both normoxia and hypoxia conditions, and (3) that of *E-cadherin* was markedly upregulated under hypoxia conditions. The mRNA expression of *HIF1α* was upregulated by hypoxia conditions compared with that in normoxia conditions, and those effects were further enhanced by TGF-β1 and -β2. The mRNA expression of *PGC1α* was markedly downregulated by TGF-β isoforms, with more potent effects of TGF-β1 and -β2 under normoxia conditions, and similar but smaller TGF-β isoform effects were observed under hypoxia conditions. The mRNA expression of *VEGF* was substantially upregulated by TGF-β1 under both normoxia and hypoxia conditions. Collectively, the results suggested that TGF-β isoforms and oxygen conditions induced diverse effects on the mRNA expression of ECM proteins and tight junction-related molecules *HIF1a* and *VEGF*, suggesting that these multiple factors may contribute to the fluctuation of physical states of the ARPE19 cell monolayer evaluated by TEER.

To study these issues further, the TGF-β isoform-induced influence on the spatial spreading of the EMT and physical aspects of 3D ARPE19 spheroids were compared, including mean areas and hardness (Figure 3, Figure 4 and Figure 5). Their stiffness, but not their mean areas, were significantly increased by only hypoxia stimulation; however, the influence of TGF-β isoforms substantially varied among the isoforms, and simultaneous conditions were observed with hypoxia. That is, (1) upon administration of TGF-β1 and TGF-β2, the 3D ARPE19 spheroids became significantly downsized and relatively stiff, and those influences were more prominent with TGF-β1; and (2) such TGF-β effects were further enhanced by hypoxia conditions.

To characterize these diverse TGF-β isoform-induced effects on 2D and 3D cultures of ARPE19 cells in more detail, the mitochondrial and glycolytic functions were studied through Seahorse cellular metabolic analysis. The results for the 2D ARPE19 cells and the ARPE 3D spheroids are shown in Figure 6 and Figure 7, respectively. Indices of mitochondrial functions were decreased by treatment with TGF-β1~3 under normoxia conditions in both 2D and 3D conditions, as shown in our previous study [20], whereas the effect of TGF-β isoforms was prominent in the 2D cells compared with the 3D spheroids. Mitochondrial respiratory functions and glycolytic functions were markedly reduced in both 2D ARPE19 cells and 3D ARPE19 spheroids under hypoxia, but the effects of TGF-β1-3 on these metabolic indices were diminished under hypoxic conditions compared with normoxia conditions. Those diverse results in these cellular metabolic functions under different oxygen conditions were rationally supported by those of the mRNA expressions of oxygen-sensitive factors *HIF1α* and *PGC1α*, as above. Collectively, TGF-β isoforms had different effects on (1) barrier functions of an ARPE19 monolayer, (2) cellular metabolic functions, and (3) spatially spreading EMT, and those effects were also modulated by hypoxia conditions.

## 4. Discussion

EMT is an indispensable biological process that occurs during various pathogenic conditions, such as fibrosis and cancer, in addition to physiological conditions, including embryonic development and wound healing [35,36]. Within the field of ophthalmology, EMT occurring in the RPE is known to be a pivotal underlying mechanism of several retinal diseases including PVR and ARMD. EMT is generally regulated by various growth factors and cytokines such as TGF-βs [37]. In fact, subretinal fibrotic lesions observed in PVR and ARMD are myofibroblasts, which are not normally present in adult tissues but are produced by EMT of the RPE [38]. In the case of ARMD, cell–cell adhesions and apical–basal polarity of RPE cells are diminished, and RPE cells thereby transform into mesenchymal cells through EMT [39]. In addition, it has been shown that so-called endothelial–mesenchymal transition (EndMT) also contributes to the cell population of the EMT [40]. In fact, in neovascular ARMD (nARMD), angiogenesis facilitates the recruitment of inflammatory cells as well as increases in oxygen supply and nutrients to the affected region [41], and neovessels thereby sprout from either the choroid or the deep retinal vessels to develop choroidal neovascular (CNV) membranes or intraretinal angiomatous proliferative lesions. These neovessels are leaky vessels and therefore induce retinal edema and hemorrhage. Therefore, the process of EMT and EndMT in the RPE is not limited to subretinal fibrosis in ARMD but rather has also been extensively studied and described in the context of proliferative vitreoretinopathy (PVR) [7], the epiretinal membrane (ERM) [8], proliferative diabetic retinopathy (PDR) [9], and the retinopathy of prematurity (ROP) [5,10]. Collectively, EMT of the RPE is an extremely important mechanism causing these vitreoretinal diseases as well as a pivotal therapeutic target. Therefore, a reliable experimental approach should be necessary to understand the underlying mechanisms and to identify factors that influence EMT of the RPE.

In fact, the RPE secretes TGF-β1 and -β2, which induce various retinal pathogeneses such as PVR and ARMD, as stated above, in addition to retinal physiology, that is, RPE cell polarity is determined by TGF-β to maintain the normal physiology of the RPE [42]. As for the contribution of TGF-β isoforms to EMT of the RPE, TGF-β1 and -β2 are predominant isoforms based on previous studies in which TGF-β1 and -β2 were detected in vitreous fluids and subretinal fluid obtained from patients with PVR [43,44]. On the other hand, recent studies have revealed that in addition to TGF-β1 and -β2, TGF-β3 was also detected in AH specimens from patients with glaucoma, suggesting that TGF-β1~3 may be differently involved in the different types of glaucoma pathogenesis [27,45,46,47,48]. In fact, we also found that TGF-β1~3 had different effects on physical and biological aspects of in vitro 2D- and 3D-cultured models generated from human trabecular meshwork (HTM) cells, which replicate the monolayer and multiple layers of the HTM [28,49]. Therefore, those observations suggested that intraocularly present TGF-β3 may also have some effects on EMT of the RPE, in addition to TGF-β1 and -β2.

Upon decreasing oxygen levels (hypoxia), physiological responses are initiated to (1) stimulate blood vessel shunt within the lung, (2) increase respiration, and (3) stimulate erythropoietin (EPO) production to increase the hemoglobin concentration in blood [50,51]. However, hypoxia also has various pathogenic effects through HIF1-mediated signaling [52,53]. HIF1 includes the subunits HIF-1α and -1β, and only HIF-1α is detected under hypoxia conditions in contrast to stable HIF-1β [54]. It has also been shown that an HIF-related mechanism is involved in the pathogenic EMT of RPE-related retinal angiogenesis such as in CNV development [55] and DR [56,57,58]. Since it was shown that HIF1α promotes the TGF-β2-related EMT of ARPE-19 cells [59], TGF-β and hypoxia are closely linked in the pathogenic EMT of the RPE. To support this, previous studies have shown that TGF-β and hypoxia stimulate the secretion of vascular endothelial growth factors (VEGFs), which act on endothelial cells to initiate and promote angiogenesis during physiological and pathological conditions [60,61], from human retinal pigment epithelium (ARPE19) cells [62]. In our preceding study, we also showed by using 2D- and 3D-cultured ARPE19 cells that TGF-β2 synergistically evoked EMT of the RPE with a hypoxia condition [20]. In the present study, we consistently observed such synergistical effects of TGF-β2 with hypoxia on EMT of the RPE. In addition, although the effects of TGF-β1 and -β3 were similar to the effects of TGF-β2, those effects were exclusively modulated by TGF-β isoforms and oxygen levels in addition to cell culture conditions of two and three dimensions, suggesting EMT of the RPE may be modulated by various pathogenic stages and the local environment. In fact, during the development of pathogenic EMT of the RPE, RPE dysfunctions are initiated as early manifestations, such as a deterioration of RPE barrier function and disorganized RPE polarization. Subsequently, EMT characteristics become evident. These include less differentiated RPE cells, upregulation of various mesenchymal cell markers, and increased migration ability. This results in the development of various degenerative retinal diseases [63,64,65,66].

It is also well known that EMT is closely associated with metabolic reprogramming. Indeed, a recent study by Cao et al. showed that EMT in RPE cells is alleviated by the preservation of mitochondrial function [67]. In the present study, hypoxia significantly decreased the mitochondrial respiratory functions of ARPE19 cells in both 2D and 3D culture conditions. Treatment with TGF-β isoforms also reduced mitochondrial spare respiratory functions in ARPE19 cells, but these effects were not markedly enhanced by hypoxia. Thus, in ARPE19 cells, hypoxia and TGF-β may reduce mitochondrial respiratory functions in part through a common pathway. Interestingly, among the TGF-β isoforms, the effects of TGF-β1 on cellular metabolic functions were most pronounced, possibly being related to the fact that barrier function and extracellular matrix gene expression were most significantly altered in the TGF-β1 treatment group. Intriguingly, the present study showed that there was no significant increase in EMT-induced compensatory glycolysis by TGF-β isoforms in both 2D and 3D culture conditions. This finding seems to be consistent with the results of a previous study by Shu et al. showing that TGF-β2-induced changes in OCR indices in RPE cells were more prominent than the increases in ECAR indices [15], suggesting that EMT in RPE cells is associated with not only metabolic remodeling through enhanced glycolysis but also with complex mechanisms including changes in fatty acid and amino acid metabolism. Overall, the results for metabolic functions in the present study suggest that reduced mitochondrial respiratory functions are prominent due to TGF-β isoform-induced EMT in RPE cells, and other processes in addition to glycolysis in EMT in the RPE should be clarified in future studies, such as fatty acid and amino acid metabolism.

As a limitation of this study, it was not determined why the TGF-β isoforms involved in EMT of the RPE are also modulated by oxygen levels (normoxia vs. hypoxia) and local environments (2D vs. 3D). In terms of a 3D culture approach related to the RPE, the aims of most studies were focused on future transplantation therapy against various retinal diseases using transplantation of human embryonic stem cells/induced pluripotent stem cells (hESCs/hiPSCs) [68,69,70]. However, our study approach using both 2D and 3D experiments in the RPE to determine various factors such as TGF-β, oxygen conditions, and other factors is quite unique and original. In fact, in addition to conventional 2D planar culture, the use of 3D spheroid culture facilitates a better understanding of molecular mechanisms of TGF-β isoforms and/or hypoxia-induced effects not only on physiological states of the RPE monolayer but also pathogenic states of multiple layers observed in proliferative vitreoretinopathy as well as CNV in ARMD. Furthermore, our results suggested potential regulatory linkages between HIF1 and TGF-β signaling in the regulation of EMT of the RPE. Therefore, further investigations using RNA-Seq analysis, specific inhibitors, and/or SiRNA against possible factors linked with TGF-β-related signaling such as Smad and non-Smad pathways [71] and HIF1β-related factors [72] will be required as our next future project.

## 5. Conclusions

In conclusion, EMT of the RPE is regulated by TGF-β isoforms (TGF-β1~3), and hypoxia has a synergistic effect with these isoforms. These interactions highlight the complexity of regulatory mechanisms in retinal diseases. A comprehensive understanding of these processes is important for retinal health.

## Figures and Tables

**Figure 1 bioengineering-11-00581-f001:**
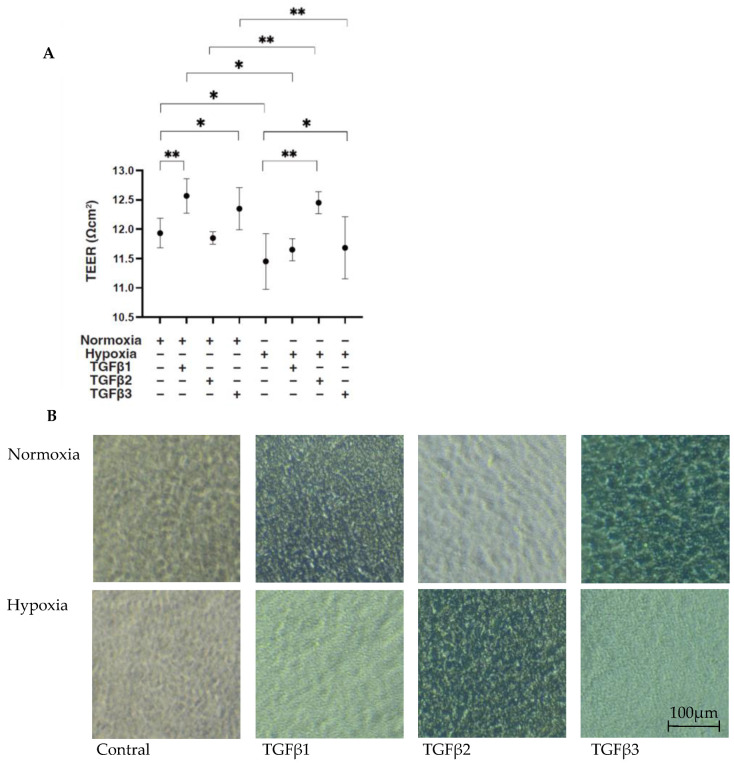
TGF-β isoform-induced effects on barrier functions of 2D ARPE19 monolayers under different oxygen conditions. Under normoxia and hypoxia conditions, nontreated 2D-cultured ARPE19 cell monolayers (NT) and 2D-cultured ARPE19 cell monolayers treated with 5 ng/mL TGF-β1~3 (TGFβ1~3) were prepared, and those culture states were observed using a phase-contrast microscope to ensure that they were monolayers under all experimental conditions. Representative phase-contrast microscopic images are shown in panel (**B**). Electric resistance (Ωcm^2^) of these 2D-cultured ARPE19 monolayers (*n* = 3 in each experimental condition) was measured using a TEER measurement device, and TEER values were plotted in panel (**A**). Scale bar (100 μm). * *p* < 0.05, ** *p* < 0.01.

**Figure 2 bioengineering-11-00581-f002:**
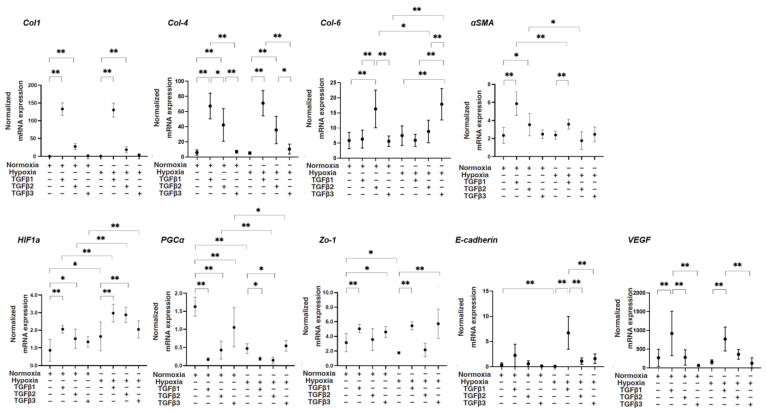
TGF-β isoform-induced effects on mRNA expression of major ECM molecules *COL1*, *COL4*, and *COL6; αSMA*; *HIF1α*; and *PGC1α*, a master regulator for mitochondrial respiration; tight junction-related molecules *ZO1* and *E-cadherin*; and *VEGF* of 2D culture of ARPE19 cells under different oxygen conditions. Under normoxia and hypoxia conditions, 2D-cultured ARPE19 cells that were not treated (NT) or treated with 5 ng/mL TGF-β1~3 (TGFβ1~3) (*n* = 5 in each experimental condition) were subjected to qPCR analysis. Levels of mRNA expression of *COL1*, *COL4*, *COL6*, *aSMA*, *HIF1a*, *PGC1a*, *ZO1*, *E-cadherin*, and *VEGF* were plotted. * *p* < 0.05, ** *p* < 0.01.

**Figure 3 bioengineering-11-00581-f003:**
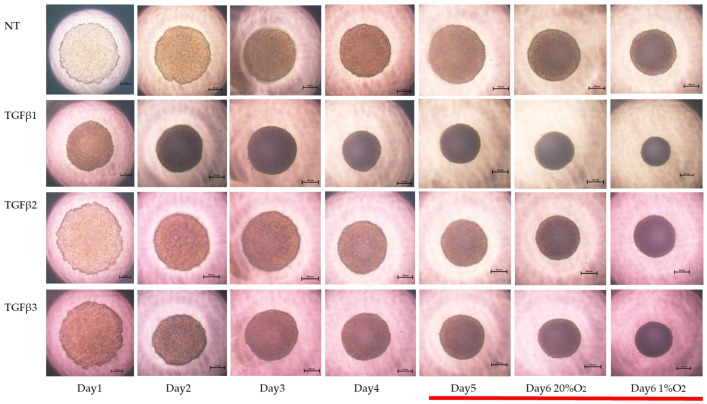
TGF-β isoform-induced effects on the appearance of 3DARPE19 spheroids under different oxygen conditions. Three-dimensional ARPE19 spheroids generated under normoxia and hypoxia conditions during 6-day spheroid culture period were treated without (NT) or with 5 ng/mL TGF-β1~3 (TGFβ1~3). Maturation processes of the 3D spheroid were observed by phase-contrast microscopy. Representative images at Days 1–6 (normoxia) and at Day 6 (hypoxia) are shown. Scale bar (100 μm).

**Figure 4 bioengineering-11-00581-f004:**
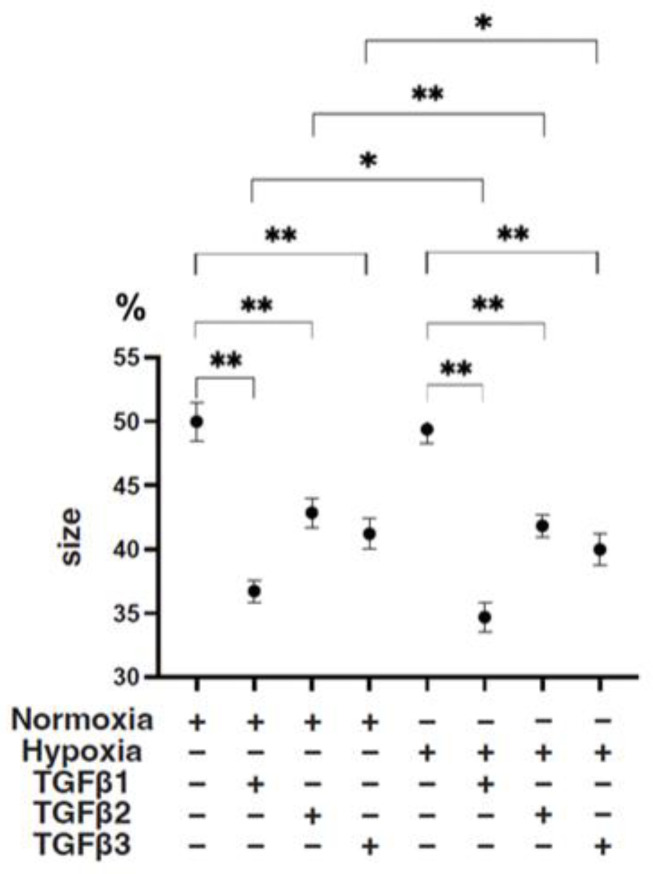
TGF-β isoform-induced effects on the mean sizes of 3D ARPE19 spheroids under different oxygen conditions. Under normoxia and hypoxia conditions, nontreated 3D ARPE19 spheroids (NT) and 3D ARPE19 spheroids treated with 5 ng/mL TGF-β1~3 (TGFβ1~3) were generated. The mean areas (μm^2^) of 3D ARPE19 spheroids were determined using phase-contrast images (*n* = 16 each), and relative values (%) against the areas of NT (50%) were plotted to more easily compare among conditions. * *p* < 0.05, ** *p* < 0.01.

**Figure 5 bioengineering-11-00581-f005:**
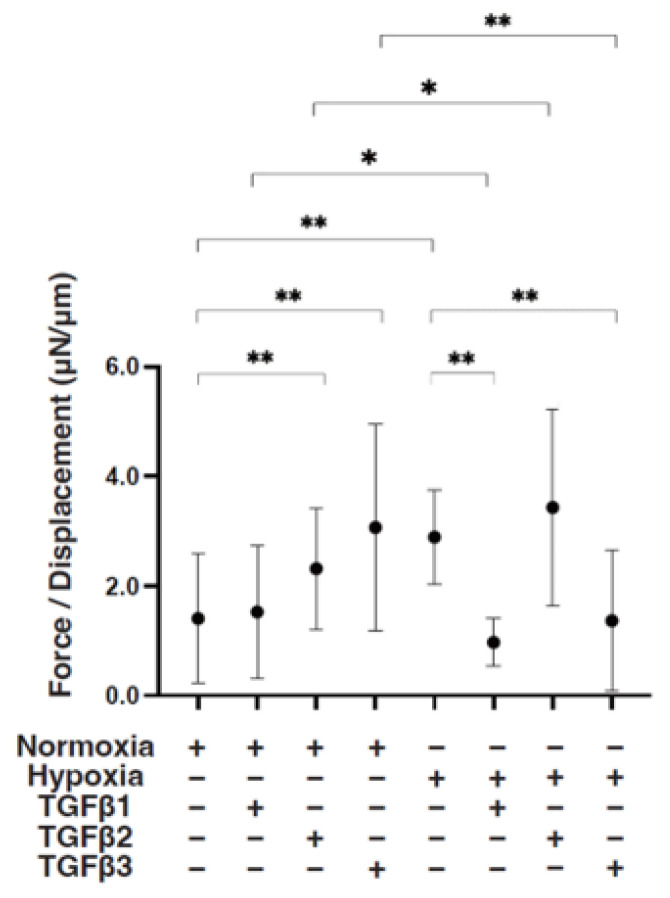
TGF-β isoform-induced effects on the stiffness of 3D ARPE19 spheroids under different oxygen conditions. Under normoxia and hypoxia conditions, nontreated 3D ARPE19 spheroids (NT) and 3D ARPE19 spheroids treated with 5 ng/mL TGF-β1~3 (TGFβ1~3) were generated. To estimate physical stiffness of the 3D ARPE19 spheroids, living spheroids (*n* = 16 each) were each directly compressed using a micro-squeezer until their semi-diameter could be measured during a period of 20 s. The required force (μN) was measured and divided by the distance of the semi-diameter. As the index represents 3D spheroid stiffness, the value of μN/μm was plotted. * *p* < 0.05, ** *p* < 0.01.

**Figure 6 bioengineering-11-00581-f006:**
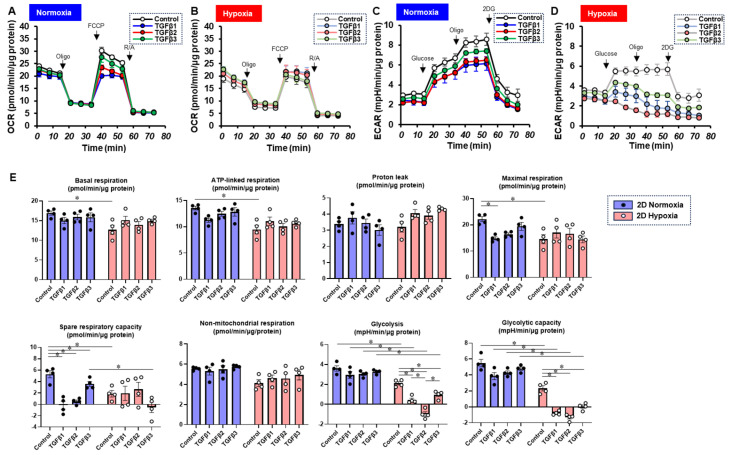
TGF-β isoform-induced effects on the mitochondrial and glycolytic functions of 2D-cultured ARPE19 cells under different oxygen conditions. Under normoxia and hypoxia conditions, nontreated 3D ARPE19 spheroids (NT) and 2D-cultured ARPE19 cells treated with 5 ng/mL TGF-β1~3 (TGFb1~3) were generated. Each specimen (*n* = 4) was subjected to a Seahorse real-time metabolic function analysis. (**A**) Oxygen consumption rate (OCR) in the Cell Mito Stress Test under normoxia (~21% O_2_) condition. (**B**) OCR in the Cell Mito Stress Test under hypoxia (~3% O_2_) condition. (**C**) Extracellular acidification rate (ECAR) in the Glycolytic Stress Test under normoxia (~21% O_2_) condition. (**D**) ECAR in the Glycolytic Stress Test under hypoxia (~3% O_2_) condition. (**E**) Indices of metabolic parameters. Basal respiration: OCR_(baseline)_–OCR_(R/A)_, ATP-linked respiration: OCR_(baseline)_–OCR_(Oligo)_, proton leak: OCR_(Oligo)_–OCR_(R/A)_, maximal respiration: OCR_(FCCP)_–OCR_(R/A)_, spare respiratory capacity: OCR_(FCCP)_–OCR_(baseline)_, non-mitochondrial respiration: OCR_(R/A)_, glycolysis: ECAR_(glucose)_–ECAR_(final measurement with 2DG)_, glycolytic capacity: ECAR_(glucose)_–ECAR_(baseline)_, oligo: oligomycin, FCCP: carbonyl cyanide p-trifluoromethoxyphenylhydrazone, Rot/AA: rotenone/antimycin A: 2-DG, 2-deoxyglucose. * *p* < 0.05.

**Figure 7 bioengineering-11-00581-f007:**
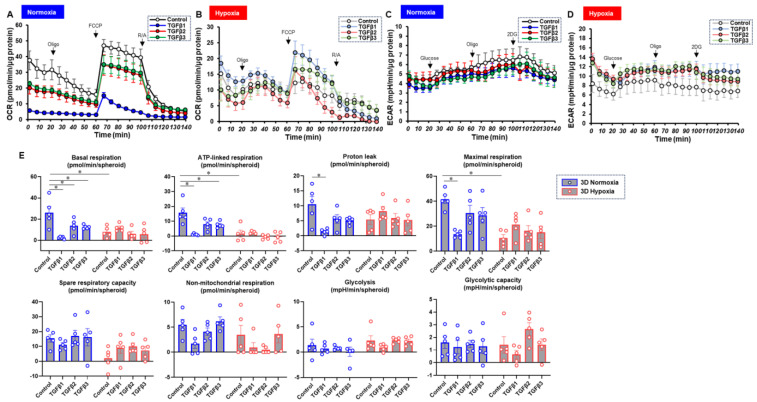
TGF-β isoform-induced effects on the mitochondrial and glycolytic functions of 3D-cultured ARPE19 cells under different oxygen conditions. Under normoxia and hypoxia conditions, nontreated 3D ARPE19 spheroids (NT) and 3D ARPE19 spheroids treated with 5 ng/mL TGF-β1~3 (TGFb1~3) were generated. Each specimen (*n* = 5) was subjected to a Seahorse real-time metabolic function analysis. (**A**) Oxygen consumption rate (OCR) in the Cell Mito Stress Test under normoxia (~21% O_2_) condition. (**B**) OCR in the Cell Mito Stress Test under hypoxia (~3% O_2_) condition. (**C**) Extracellular acidification rate (ECAR) in the Glycolytic Stress Test under normoxia (~21% O_2_) condition. (**D**) ECAR in the Glycolytic Stress Test under hypoxia (~3% O_2_) condition. (**E**) Indices of metabolic parameters. Basal respiration: OCR_(baseline)_–OCR_(R/A)_, ATP-linked respiration: OCR_(baseline)_–OCR_(Oligo)_, proton leak: OCR_(Oligo)_–OCR_(R/A)_, maximal respiration: OCR_(FCCP)_–OCR_(R/A)_, spare respiratory capacity: OCR_(FCCP)_–OCR_(baseline)_, non-mitochondrial respiration: OCR_(R/A)_, glycolysis: ECAR_(glucose)_–ECAR_(final measurement with 2DG)_, glycolytic capacity: ECAR_(glucose)_–ECAR_(baseline)_, oligo: oligomycin, FCCP: carbonyl cyanide p-trifluoromethoxyphenylhydrazone, Rot/AA: rotenone/antimycin A: 2-DG, 2-deoxyglucose. * *p* < 0.05.

## Data Availability

The original contributions presented in the study are included in the article/Appendix A, further inquiries can be directed to the corresponding author.

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
