# Peer review of "TGF-β Isoforms and Local Environments Greatly Modulate Biological Nature of Human Retinal Pigment Epithelium Cells"

_bioengineering, 2024, doi:10.3390/bioengineering11060581_

Round 1
Reviewer 1 Report
Comments and Suggestions for Authors
In this manuscript, Nshikiori et al explore the effects of TGF beta factor isoform on transformed ARPE19 cell line under normoxia and hypoxia conditions. The TEER measurements, RNA analysis of a few genes via qPCR, and metabolic analysis by analyzing mitochondrial respiration and glycolytic rates via Seahorse assay were performed. The results suggest that some isoforms affect TEER values, RNA expression and metabolic activity of these cells.
ARPE19 is considered to be a spontaneously transformed cell line and may not recapitulate appropriate morphology relevant to RPE unless grown under proper conditions. Considering this, it is important to include brightfield images of the ARPE19 cells (2D) under normoxia and hypoxia conditions for each of the treatments. Please include brightfield images of 2D cultures similar to as shown for 3D cultures in Figure 3.
RPE is an epithelial cell type that grows as a monolayer. Please explain in the manuscript why 3D cultures were used for assessment and how s it relevant to the cell type.
Lines 324-326: “To support this, previous studies have 324 shown that TGF-β and hypoxia stimulate secretion of vascular endothelial growth factors (VEGFs), which act on endothelial cells to initiate and promote angiogenesis during physiological and pathological conditions”. Since the TGF beta factors affect VEGF secretion, was qPCR performed for VEGF as well? For this claim, it will be important to show how VEGF RNA or protein levels change with exposure to different TGF beta isoforms.
It appears that the isoforms that increase OCR, also increase ECAR. Please include in discussion how that is relevant, and how metabolic changes with TGF beta isoforms are relevant to diseases.
Lines 187-189: “The levels of mRNA expression of ZO1 were almost identical to the TEER values in the above-described conditions, supporting the data of TGF-β isoforms-induced effects on barrier functions evaluated by TEER”. How could the levels of mRNA compare in value with TEER? Both may increase or decrease similarly, but stating that mRNA levels were identical to TEER values is not correct and would not mean anything.
In Methods, please explain how RNA was extracted and cDNA prepared, and what was data normalized to. Is the data presented as Cq values? Please clarify.
Lines 205-206: Please correct the sentence “Under normoxia and hypoxia conditions, nontreated the 2D cultured ARPE19 cell monolayers (NT) and the 2D cultured ARPE19 cells…..”
Comments on the Quality of English LanguageQuality of English language is good
Author Response
Dear Editor,
Thank you very much for the constructive comments concerning our manuscript “TGF-b isoforms and local environments greatly modulate biological natures of human retinal pigment epithelium cells”. We carefully checked all of the Editor and Reviewer comments and prepared a revised version of our paper that takes these comments into account. The changes are listed below.
Reviewer comments:
Reviewer 1
In this manuscript, Nshikiori et al explore the effects of TGF beta factor isoform on transformed ARPE19 cell line under normoxia and hypoxia conditions. The TEER measurements, RNA analysis of a few genes via qPCR, and metabolic analysis by analyzing mitochondrial respiration and glycolytic rates via Seahorse assay were performed. The results suggest that some isoforms affect TEER values, RNA expression and metabolic activity of these cells.
- ARPE19 is considered to be a spontaneously transformed cell line and may not recapitulate appropriate morphology relevant to RPE unless grown under proper conditions. Considering this, it is important to include brightfield images of the ARPE19 cells (2D) under normoxia and hypoxia conditions for each of the treatments. Please include brightfield images of 2D cultures similar to as shown for 3D cultures in Figure 3.
Answer; We sincerely appreciate your excellent comment. As suggested, representative phase contrast images of 2D cultured cells were included in Fig. 1.
- RPE is an epithelial cell type that grows as a monolayer. Please explain in the manuscript why 3D cultures were used for assessment and how is it relevant to the cell type.
Answer; We sincerely appreciate your excellent comment. In terms of use of 3D cell culture for an epithelial phenotype RPE, we would like to evaluate effects of TGF-b isoforms and hypoxia on physiological state of RPE monolayer as well as pathogenic state of multiple layers observed in proliferative vitreoretinopathy as well as CNV in ARMD. Therefore, these information are included in the last paragraph of Discussion: “As a limitation of this study, it has not been determined why the TGF-b isoforms on EMT of the RPE are also modulated by oxygen levels, normoxia vs hypoxia and local environments, 2D vs 3D. In terms of 3D culture approach related to RPE, most of studies were aimed for future transplantation therapy against various retinal diseases using human embryonic stem cells/induced pluripotent stem cells (hESCs/hiPSCs) [64][65][66]. However, our study approach using both 2D and 3D experiments in RPE to determine various factors such as TGF-b, oxygen conditions and others are quite unique and original. In fact, in addition to conventional 2D planar culture, the use of 3D spheroid culture facilitates better understanding of molecular mechanisms of TGF-b isoforms and/or hypoxia-induced effects on not only physiological states of RPE monolayer but also pathogenic states of multiple layers observed in proliferative vitreoretinopathy as well as CNV in ARMD. Furthermore, our current results suggested that potential regulatory linkages between HIF1 and TGF-b signaling in the regulation of EMT of the RPE. Therefore, further investigations using RNA-Seq analysis, specific inhibitors and/or SiRNA against possible factors linked with TGF-b-related signaling such as Smad and non-Smad pathways [67], and HIF1a-related factors [68] will be required as our next future project.”.
- Lines 324-326: “To support this, previous studies have 324 shown that TGF-β and hypoxia stimulate secretion of vascular endothelial growth factors (VEGFs), which act on endothelial cells to initiate and promote angiogenesis during physiological and pathological conditions”. Since the TGF beta factors affect VEGF secretion, was qPCR performed for VEGF as well? For this claim, it will be important to show how VEGF RNA or protein levels change with exposure to different TGF beta isoforms.
Answer; We sincerely appreciate your excellent comment. As suggested, qPCR for VEGF was included in Fig. 2.
- It appears that the isoforms that increase OCR, also increase ECAR. Please include in discussion how that is relevant, and how metabolic changes with TGF beta isoforms are relevant to diseases.
Answer; We sincerely appreciate your excellent suggestions. According to this opinion, we have included a new paragraph mentioning the metabolic effect on TGF-β isoforms induced EMT in the revised discussion section (3rd paragraph).
- Lines 187-189: “The levels of mRNA expression of ZO1 were almost identical to the TEER values in the above-described conditions, supporting the data of TGF-β isoforms-induced effects on barrier functions evaluated by TEER”. How could the levels of mRNA compare in value with TEER? Both may increase or decrease similarly, but stating that mRNA levels were identical to TEER values is not correct and would not mean anything.
Answer; We sincerely appreciate your excellent comment. I agree that changes in TEER by various conditions should not be explained by only mRNA expression of ZO1, and our corresponding statement was definitely overstated. As suggested by both reviewers, additional qPCR analysis of other related molecules including COL4, COL6, VEGF and E-cadherin was required. Therefore, those data were included in Fig. 2 and corresponding result was rewritten: “Next, levels of mRNA expression of various molecules including major ECM proteins, collagen 1 (COL1), COL4 and COL6 and a-smooth muscle actin (aSMA), hypoxia-inducible factor 1a (HIF1a), PGC1a, a master regulator for mitochondrial respiration, tight junction-related molecules, ZO1 and E-cadherin, and VEGF were compared among conditions as described above. As shown in Fig. 2, despite the fact that mRNA expression of COL1 and COL4 was not altered by TGF-b3, their expression was substantially upregulation by TGF-b1 and TGF-b2, and those increase levels of TGF-b1 were much more potent under both normoxia and hypoxia conditions. Similarly to this, gene expression of aSMA was also substantially upregulated by TGF-b1 as compared to others under both different oxygen conditions. However, in contrast, mRNA expression of COL6 was marked up-regulated by TGF-b3 under both normoxia and hypoxia conditions. As for the levels of mRNA expression of tight junction related molecules, 1) those of ZO1 and E-cadherin were significantly down-regulated by hypoxia conditions, 2) those of ZO1 were substantially up-regulated by TGF-b1 and TGF-b3 under both normoxia and hypoxia conditions, those of E-cadherin were markedly up-regulated under hypoxia conditions. The mRNA expression of HIF1a was upregulated by hypoxia conditions compared with that by normoxia conditions, and those effects were further enhanced by TGF-b1 and -b2. The mRNA expression levels of PGC1a were markedly downregulated by TGF-b isoforms with more potent effect by TGF-b1 and -b2 under normoxia condition, and similar but lesser TGF-b isoform induced effects were observed under hypoxia conditions. The mRNA expression of VEGF was substantially up-regulated by TGF-b1 under both normoxia and hypoxia conditions. Collectively, these results suggested that TGF-b isoforms and oxygen conditions induced diverse effects on mRNA expression of ECM proteins, tight-junction related molecules, HIF1a and VEGF, suggesting that these multiple factors may contribute fluctuation of physical states of the ARPE19 monolayer evaluated by TEER.”.
- In Methods, please explain how RNA was extracted and cDNA prepared, and what was data normalized to. Is the data presented as Cq values? Please clarify.
Answer; We sincerely appreciate your excellent comment. Detail of methods for qPCR was included in the corresponding Method: “Using specific primers and probes as shown in supplemental Table 1, qPCR analysis of the 2D or 3D cultures of ARPE19 cells was performed as previously reported [17,34]. Briefly, total RNA was extracted using an RNeasy mini kit (Qiagen, Valencia, CA) and reverse transcription was processed with the SuperScript IV kit (Invitrogen). Respective gene expression was quantified by real-time PCR (Cq value=40) with the Taqman Master mix (Applied Biosystems/Thermo Fisher Scientific).”.
- Lines 205-206: Please correct the sentence “Under normoxia and hypoxia conditions, nontreated the 2D cultured ARPE19 cell monolayers (NT) and the 2D cultured ARPE19 cells…..”
Answer; We sincerely appreciate your excellent comment. As pointed out, this sentence was corrected; “Under normoxia and hypoxia conditions, 2D cultured ARPE19 cell were untreated (NT) and treated with 5 ng/ml TGF-β1~3 (TGFb1~3) (n=5 in each experimental condition) were subjected to qPCR analysis.”.
Reviewer 2 Report
Comments and Suggestions for Authors
This is an interesting study; the authors find that the TGF-β-induced EMT in RPE cell in the vitro model is regulated by the specific TGF-β isoforms and oxygen levels. however, a major revision is required before considering publishing the manuscript in the journal:
1. Please providing a concise brief overview of the pathogenesis of epithelial-mesenchymal transition in RPE, highlight key studies and findings, specifically addressing what has been accomplished in the research of RPE EMT, and identifying gaps in the research area which you want to fill.
2. In line 16-17, the authors describe "qPCR analysis of major ECM components including collagen 1 (COL1) and α smooth muscle actin (αSMA)." This important to consider whether this concept is accurate. Muscle alpha-actin, including αSMA, is a protein component of the muscle thin filament. Technically, it does not belong to the extracellular matrix (ECM). Its functions primarily involve cell motility, structure, integrity, intercellular signaling, and responsiveness to external stimuli presented by the ECM, such as collagens, elastin, and fibronectin.
3. It's widely acknowledged that E-cadherin is downregulated in EMT RPE cells, which is a major feature of EMT formation. It would be beneficial for the authors to include the results of E-cadherin expression in the study.
4. Explain why both 2D and 3D models are crucial in studying RPE EMT.
5. Describe how do you validate the formation of RPE cells monolayers in your study.
6. Provide detailed information about the approach of both 2D and 3D experiments in RPE.
Clarify the relationship between hypoxia-inducible factor 1 (HIF1), vascular endothelial growth factor, and transforming growth factor-beta in RPE EMT. Address whether HIF1 directly stimulates RPE EMT or acts through TGF-beta signaling? and provide evidence or discuss any potential regulatory interactions between HIF1 and TGF-beta in the regulation of RPE EMT.
Comments on the Quality of English Language1Please proofread the text to minimize grammatical errors.
Author Response
Dear Editor,
Thank you very much for the constructive comments concerning our manuscript “TGF-b isoforms and local environments greatly modulate biological natures of human retinal pigment epithelium cells”. We carefully checked all of the Editor and Reviewer comments and prepared a revised version of our paper that takes these comments into account. The changes are listed below.
Reviewer comments:
Reviewer 2
This is an interesting study; the authors find that the TGF-β-induced EMT in RPE cell in the vitro model is regulated by the specific TGF-β isoforms and oxygen levels. however, a major revision is required before considering publishing the manuscript in the journal:
- Please providing a concise brief overview of the pathogenesis of epithelial-mesenchymal transition in RPE, highlight key studies and findings, specifically addressing what has been accomplished in the research of RPE EMT, and identifying gaps in the research area which you want to fill.
Answer; We sincerely appreciate your excellent comment. As suggested, a concise brief overview of the pathogenesis of epithelial-mesenchymal transition in RPE, highlight key studies and findings was included 1st paragraph of Discussion: “EMT is an indispensable biological process that occurs during various pathogenic conditions, such as fibrosis and cancer, in addition to physiological conditions, including embryonic development and wound healing [35,36]. Within the field of ophthalmology, EMT occurring in the RPE is known to be a pivotal underlying mechanism of several retinal diseases including PVR and ARMD. EMT is generally regulated by various growth factors and cytokines such as TGF-βs [37]. In fact, subretinal fibrotic lesions observed in PVR and ARMD are myofibroblasts, which are not normally present in adult tissues but are produced by EMT of the RPE [38]. In the case of ARMD, cell–cell adhesions and apical–basal polarity of RPE cells are diminished, thereby RPE cells transform into mesenchymal cells through EMT [39]. In addition, it is shown that so called “endothelial–mesenchymal transition (EndMT”) also contribute in cell population the EMT [40]. In fact, in neovascular ARMD (nARMD), angiogenesis facilitate recruitment of inflammatory cells as well as increase of oxygen supply and nutrients to the affected region [41], thereby neovessels sprout from either the choroid or the deep retinal vessels to develop choroidal neovascular membranes (CNV) or intraretinal angiomatous proliferative lesions. These neovessels are leaky vessels and therefore induce to retinal edema and hemorrhage. Therefore, the process of EMT and EndMT in RPE is not limited to subretinal fibrosis in AMD but rather, has also been extensively studied and described in the context of proliferative vitreoretinopathy (PVR) [7], epiretinal membrane (ERM) [8], proliferative diabetic retinopathy (PDR) [9]. proliferative diabetic retinopathy (PDR) and retinopathy of prematurity (ROP) [5,10]. Collectively, EMT of RPE is an extremely important mechanisms causing these vitreoretinal diseases as well as the pivotal therapeutic target. Therefore, a reliable experimental approach should be necessary to understand underlying mechanisms and to identify influence factors of the EPM of the RPE.”.
- In line 16-17, the authors describe "qPCR analysis of major ECM components including collagen 1 (COL1) and α smooth muscle actin (αSMA)." This important to consider whether this concept is accurate. Muscle alpha-actin, including αSMA, is a protein component of the muscle thin filament. Technically, it does not belong to the extracellular matrix (ECM). Its functions primarily involve cell motility, structure, integrity, intercellular signaling, and responsiveness to external stimuli presented by the ECM, such as collagens, elastin, and fibronectin.
Answer; We sincerely appreciate your excellent comment. As suggested, qPCR for COL4 and COL6 was included in Fig. 2.
- It's widely acknowledged that E-cadherin is downregulated in EMT RPE cells, which is a major feature of EMT formation. It would be beneficial for the authors to include the results of E-cadherin expression in the study.
Answer; We sincerely appreciate your excellent comment. As suggested, qPCR for E-cadherin was included in Fig. 2.
- Explain why both 2D and 3D models are crucial in studying RPE EMT.
Answer; We sincerely appreciate your excellent comment. In terms of use of 3D cell culture for an epithelial phenotype RPE, we would like to evaluate effects of TGF-b isoforms and hypoxia on physiological state of RPE monolayer as well as pathogenic state of multiple layers observed in proliferative vitreoretinopathy as well as CNV in ARMD.
- Describe how do you validate the formation of RPE cells monolayers in your study.
Answer; We sincerely appreciate your excellent comment. In terms of validation of the formation of RPE cells monolayers, we confirmed this issue by phase contrast microscopy observation, and therefore representative phase contrast images of 2D cultured cells were included in Fig. 1,
- Provide detailed information about the approach of both 2D and 3D experiments in RPE.
Answer; We sincerely appreciate your excellent comment. As far as I survey, 3D culture approach related to RPE, using human embryonic stem cells/induced pluripotent stem cells (hESCs/hiPSCs), most of studies were aimed for future transplantation therapy against various retinal diseases (PMID: 27826002, PMID: 36963105, PMID: 38229139), and therefore, our approach of both 2D and 3D experiments in RPE to determine various factors such as TGF-b, oxygen conditions and others are quite unique. Therefore, these information are included in the last paragraph of Discussion: “As a limitation of this study, it has not been determined why the TGF-b isoforms on EMT of the RPE are also modulated by oxygen levels, normoxia vs hypoxia and local environments, 2D vs 3D. In terms of 3D culture approach related to RPE, most of studies were aimed for future transplantation therapy against various retinal diseases using human embryonic stem cells/induced pluripotent stem cells (hESCs/hiPSCs) [64][65][66]. However, our study approach using both 2D and 3D experiments in RPE to determine various factors such as TGF-b, oxygen conditions and others are quite unique and original. In fact, in addition to conventional 2D planar culture, the use of 3D spheroid culture facilitates better understanding of molecular mechanisms of TGF-b isoforms and/or hypoxia-induced effects on not only physiological states of RPE monolayer but also pathogenic states of multiple layers observed in proliferative vitreoretinopathy as well as CNV in ARMD. Furthermore, our current results suggested that potential regulatory linkages between HIF1 and TGF-b signaling in the regulation of EMT of the RPE. Therefore, further investigations using RNA-Seq analysis, specific inhibitors and/or SiRNA against possible factors linked with TGF-b-related signaling such as Smad and non-Smad pathways [67], and HIF1a-related factors [68] will be required as our next future project.”.
- Clarify the relationship between hypoxia-inducible factor 1 (HIF1), vascular endothelial growth factor, and transforming growth factor-beta in RPE EMT. Address whether HIF1 directly stimulates RPE EMT or acts through TGF-beta signaling? and provide evidence or discuss any potential regulatory interactions between HIF1 and TGF-beta in the regulation of RPE EMT.
Answer; We sincerely appreciate your excellent comment and suggestion. As far as I survey, definite evidence of direct linkage among these factors in RPE EMT. However, as suggested, our current study may suggest potential regulatory interactions between HIF1 and TGF-beta in the regulation of RPE EMT. Therefore, this information is included in the study limitation in Discussion: “As a limitation of this study, it has not been determined why the TGF-b isoforms on EMT of the RPE are also modulated by oxygen levels, normoxia vs hypoxia and local environments, 2D vs 3D. In terms of 3D culture approach related to RPE, most of studies were aimed for future transplantation therapy against various retinal diseases using human embryonic stem cells/induced pluripotent stem cells (hESCs/hiPSCs) [64][65][66]. However, our study approach using both 2D and 3D experiments in RPE to determine various factors such as TGF-b, oxygen conditions and others are quite unique and original. In fact, in addition to conventional 2D planar culture, the use of 3D spheroid culture facilitates better understanding of molecular mechanisms of TGF-b isoforms and/or hypoxia-induced effects on not only physiological states of RPE monolayer but also pathogenic states of multiple layers observed in proliferative vitreoretinopathy as well as CNV in ARMD. Furthermore, our current results suggested that potential regulatory linkages between HIF1 and TGF-b signaling in the regulation of EMT of the RPE. Therefore, further investigations using RNA-Seq analysis, specific inhibitors and/or SiRNA against possible factors linked with TGF-b-related signaling such as Smad and non-Smad pathways [67], and HIF1a-related factors [68] will be required as our next future pr
Round 2
Reviewer 1 Report
Comments and Suggestions for Authors
The updates to the manuscript are appropriate. Pleas e include better and higher resolution images in Figure 1, as the morphology is not clear in the images shared.
Comments on the Quality of English LanguageLooks good. Please check some sentences for grammar.
Author Response
Dear Editor,
Thank you very much for the constructive comments concerning our manuscript “TGF-b isoforms and local environments greatly modulate biological natures of human retinal pigment epithelium cells”. We carefully checked all of the Editor and Reviewer comments and prepared a revised version of our paper that takes these comments into account. The changes are listed below.
Reviewer comments:
Reviewer 1
We sincerely appreciate your excellent comment.
- Please include better and higher resolution images in Figure 1, as the morphology is not clear in the images shared.
Answer; I modified the images in Figure 1 to show the morphology more clearly.
- Please check some sentences for grammar.
Answer; I have sent the paper for English proofreading.

Reviewer 2 Report
Comments and Suggestions for Authors
The author has answered all previously raised questions and made reasonable revisions. I don't have concern.
Author Response
Dear Editor,
Thank you very much for the constructive comments concerning our manuscript “TGF-b isoforms and local environments greatly modulate biological natures of human retinal pigment epithelium cells”. We carefully checked all of the Editor and Reviewer comments and prepared a revised version of our paper that takes these comments into account. The changes are listed below.
Reviewer comments:
Reviewer 2
The author has answered all previously raised questions and made reasonable revisions. I don't have concern.
Answer; We sincerely appreciate your excellent comments.
Thank you for your kind response.